# The Tubulin Superfamily in Apicomplexan Parasites

**DOI:** 10.3390/microorganisms11030706

**Published:** 2023-03-09

**Authors:** Naomi Morrissette, Izra Abbaali, Chandra Ramakrishnan, Adrian B. Hehl

**Affiliations:** 1Molecular Biology and Biochemistry, University of California Irvine, Irvine, CA 92697, USA; 2Institute for Parasitology, University of Zurich, Winterthurerstrasse 266a, 8057 Zürich, Switzerland

**Keywords:** *Babesia*, *Besnoitia*, *Cryptosporidium*, *Cyclospora*, *Cystoisospora*, *Plasmodium*, *Sarcocystis*, *Theileria*, *Toxoplasma*, tubulin signature

## Abstract

Microtubules and specialized microtubule-containing structures are assembled from tubulins, an ancient superfamily of essential eukaryotic proteins. Here, we use bioinformatic approaches to analyze features of tubulins in organisms from the phylum Apicomplexa. Apicomplexans are protozoan parasites that cause a variety of human and animal infectious diseases. Individual species harbor one to four genes each for α- and β-tubulin isotypes. These may specify highly similar proteins, suggesting functional redundancy, or exhibit key differences, consistent with specialized roles. Some, but not all apicomplexans harbor genes for δ- and ε-tubulins, which are found in organisms that construct appendage-containing basal bodies. Critical roles for apicomplexan δ- and ε-tubulin are likely to be limited to microgametes, consistent with a restricted requirement for flagella in a single developmental stage. Sequence divergence or the loss of δ- and ε-tubulin genes in other apicomplexans appears to be associated with diminished requirements for centrioles, basal bodies, and axonemes. Finally, because spindle microtubules and flagellar structures have been proposed as targets for anti-parasitic therapies and transmission-blocking strategies, we discuss these ideas in the context of tubulin-based structures and tubulin superfamily properties.

## 1. Introduction

Microtubules are universal and essential components of eukaryotic cells that are indispensable for a variety of cellular processes, particularly mitosis, but also motility, cell shape, and intracellular trafficking. These filamentous polymers assemble from α-β-tubulin heterodimer subunits in a process that is regulated by GTP binding and hydrolysis [1,2]. The biosynthesis of functional heterodimers requires CCT and tubulin cofactors to coordinate the stable association of α- and β-tubulins, each of which binds to one molecule of GTP [3,4]. The polymerization of α-β heterodimers stimulates GTP hydrolysis in β-tubulin, causing a conformational change that weakens subunit interactions to permit microtubule disassembly. Disassembled heterodimers must exchange GDP for GTP to restore assembly competency. A dynamic equilibrium between free dimers and polymers is required for essential processes, including cell division.

α- and β-tubulins are members of a larger superfamily that consists of six subgroups (Figure 1). Members contain a tubulin signature motif ([S/A/G]-G-G-T-G-[S/A]-G) which coordinates binding to the phosphate portion of GTP [2]. Beyond this motif, sequence similarity and overall structural organization are conserved both within and between families from diverse eukaryotic lineages because functional constraints restrict variation [5]. All eukaryotes harbor genes for α- and β-tubulins to form heterodimer subunits which assemble into microtubules [5,6,7,8]. A third essential tubulin, γ-tubulin, complexes with additional proteins to form the γ-tubulin ring complex (γ-TuRC), which nucleates most microtubules [9,10]. The tubulin superfamily is thought to have arisen from gene duplication, with subsequent diversification such that tubulins in individual families carry out non-redundant functions. Tubulins arose near to the appearance of the last common eukaryotic ancestor, and α-, β-, and γ- families are found in all eukaryotes [5,11]. Gene expansion within these families led to multiple genes for α-, β- and/or γ-tubulins, known as isoforms or isotypes [12,13,14]. Isotypes are typically distinguished by minor amino acid differences that modify biochemical properties to fine-tune temporal, structural, or cell-specific requirements [12,13]. Eukaryotes have one to two genes to specify γ-tubulins and may have from one to seven or more genes that encode α- and β- tubulin isotypes [15,16,17]. Unicellular eukaryotes employ fewer α- and β- isotypes, typically specified by one to three genes.

Three other tubulins (δ-, ε-, and ζ-, known as the ZED module) contribute to centriole and basal body structures. Sequence conservation is weaker for ZED-tubulins, possibly due to lower functional constraints in these families, which are only found in some eukaryotes. Bioinformatic analysis links the function of ε-tubulin to δ- or ζ-tubulin; organisms harboring a gene for ε-tubulin also carry δ- and/or ζ-tubulin gene(s), and a lack of ε-tubulin signifies that δ- and ζ-tubulin genes will also be absent [18]. This likely reflects overlapping functions of δ- and ζ-tubulin, which are thought to be evolutionarily exchangeable [5,18]. ζ-tubulin genes are not found in apicomplexan genomes, although they are present in *Leishmania* and *Trypanosoma* species, which are in a distinct clade of protozoan parasites [6,18]. Genes for δ- and ε-tubulins have been eliminated from diverse eukaryotic lineages several times [5,19,20]. Loss is associated with an inability to form appendage-containing basal bodies, or to build basal bodies and motile flagella [20,21,22,23].

Apicomplexan parasites are a diverse group of protozoan pathogens that cause illness and death in humans and in animals, including agriculturally significant livestock. While some apicomplexans are transmitted by the bite of infected mosquito (*Plasmodium*) or tick (*Babesia* and *Theileria*) vectors, others are spread by the consumption of food or water harboring tissue cyst or oocyst stages (*Toxoplasma*, *Cryptosporidium*, *Cyclospora*, *Cystoisospora*). Perhaps the best-known apicomplexans are the *Plasmodium* species, which are categorized within the class Aconoidasida, order Haemosporida. *Plasmodium* species are the agents of human malaria, a deadly infection that is estimated to kill more than half a million individuals each year [24]. The Aconoidasida also includes the order Piroplasmida, containing *Babesia* and *Theileria* species. *Babesia* are important pathogens of cattle and emerging zoonotic infections of humans in many countries [25,26]. Bovine, ovine, and equine infections with *Theileria* cause significant livestock losses in endemic (tropical) areas [27,28]. The second class within the Apicomplexa, the Conoidasida, contains organisms that harbor a distinct tubulin-containing organelle, the conoid, which is proximal to apical secretory organelles and associated structures that give the phylum Apicomplexa its name. The Conoidasida includes orally infectious parasites in the order Eucoccidiorida, commonly known as coccidian parasites. Among these, parasites in the family Cryptosporidiidae (genus *Cryptosporidium*) are believed to represent an early branching lineage and have unusually streamlined genomes and reduced organelles relative to other apicomplexans [29]. *Cryptosporidium* resides in the intestinal epithelium and causes watery diarrhea and weight loss, resulting in serious and chronic complications in children and immunocompromised individuals [30]. *Cystoisospora* (formerly *Isospora*) and *Cyclospora* are zoonotic parasites of the Sarcocystidae and Eimeriidae families that cause serious diarrheal infections in immunocompromised individuals [31,32,33]. *Besnoitia* species (Sarcocystidae) cause pathology in grazing livestock and wildlife, and *Eimeria* species (Eimeriidae) infect poultry and livestock. Both are responsible for economic losses to the agricultural industry [34,35,36]. Lastly, *Toxoplasma gondii* (Sarcocystidae) is a globally distributed infection of all warm-blooded mammals, ranked by the Centers for Disease Control in the top five neglected infectious diseases in the US [37]. While infection can be asymptomatic in healthy human hosts, *T. gondii* causes birth defects, encephalitis, and blindness [38]. Moreover, because immunity controls but does not eliminate *Toxoplasma*, individuals remain at risk for the reactivation of latent parasites years after initial infection, particularly if immunity is compromised.

Apicomplexan parasites have complex life cycles that typically consist of asexual (vegetative) proliferation of zoite stages and genome reassortment during a transient sexual cycle. All parasite forms require spindle microtubules to coordinate chromosome segregation. In addition, zoites use cortical (subpellicular) microtubules to establish shape, rigidity, and polarity of the parasite pellicle. Subpellicular microtubules are non-dynamic and are unusually stable to detergent extraction. In *Toxoplasma*, this stability is conferred by a complex of microtubule inner proteins and is, therefore, extrinsic to tubulin [39,40]. Conoidasida class zoites also assemble the conoid, a distinctive tubulin-containing structure that incorporates repurposed proteins from the flagellar apparatus [41,42]. Importantly, apicomplexan zoites lack flagella, an organelle that underlies motility in other alveolate lineages. In *Plasmodium* and *Toxoplasma* zoites, motility has been shown to require posterior-directed capping of surface adhesins, which is driven by actin and myosin activity [43]. Although members of the Aconoidasida lack an obvious conoid, recent studies have identified conoid components in *Plasmodium* zoites, indicating the persistence of a less distinct structure [44]. In *Toxoplasma*, the conoid has been shown to bind to a formin that stimulates actin polymerization to activate gliding motility [45]. Microgametes (male) of some apicomplexans employ two flagella for motility required to fertilize macrogametes (female). Additional microtubules have been detected in electron microscopy studies of gametes, but little is known of the components or properties of these populations.

In this paper, we analyze the features of tubulins encoded by key apicomplexan organisms. We specifically consider the potential redundancy or specialization of α- and β-tubulin isotypes, the presence, absence, or divergence of δ- or ε-tubulins, and the stage-specific expression of tubulin family members. Ultimately, the properties of the tubulin superfamily inform proposed strategies to control apicomplexan parasites via tubulin-targeting antimitotic agents, or through transmission-blocking strategies that disrupt microgamete flagella [46,47,48,49,50,51].

## 2. Materials and Methods

### 2.1. Identification of Apicomplexan Tubulin Genes

Representative sequences for tubulin family members in the unicellular green alga *Chlamydomonas reinhardtii* and the ciliate *Tetrahymena thermophila* (deposited at UniProt: https://www.uniprot.org/, entries confirmed January 2023) were used to identify predicted homologs from VEuPathDB.org [52]. These organisms have well-defined members for each tubulin family and are grouped as Diaphoretickes along with apicomplexans [53]. BlastP searches [54] were used to collect all predicted tubulins from genomes of the apicomplexans *Babesia microti, Besnoitia besnoiti, Cryptosporidium parvum, Cyclospora cayetanensis, Cystoisospora suis, Eimeria tenella, Hammondia hammondi, Neospora caninum, Plasmodium falciparum, Plasmodium vivax, Sarcocystis neurona, Theileria annulata* and *Toxoplasma gondii*. Searches and analysis were initially limited to reference strains for these key pathogens. In the case that specific tubulin family members appeared to be missing, BlastP searches were expanded to encompass all sequenced genera and strains to assess whether this absence was a consistent feature or could be due to incomplete genome representation. To assign apicomplexan tubulins to specific families or to confirm existing annotations, sequences were evaluated relative to α-, β-, γ-, δ-, and ε-tubulins from *C. reinhardtii* and *T. thermophila*. Briefly, representative tubulin family members from *C. reinhardtii* and *T. thermophila* (Appendix A) were aligned with candidate apicomplexan tubulins using Clustal Omega [55,56,57]. The associated guide and phylogenetic trees were used to designate or confirm apicomplexan tubulins as members of specific families by proximity to individual *Chlamydomonas* and *Tetrahymena* tubulins.

### 2.2. Tubulin Protein Analysis

Predicted amino acid sequences for tubulins were aligned in Clustal Omega [55,56], and EMBOSS was used to determine pairwise conservation between tubulins [58]. In cases where tubulin genes are apparently mis-annotated, such as when they appear to lack typical regions or have extra sequences inserted or appended to conserved sequences, we have documented these in Table 1 and Table 2 notes and our observations have been added to VEuPathDB.org annotations.

### 2.3. Tubulin Transcriptomics and Proteomics

Relevant unique, sense strand tubulin transcript counts were extracted from several stage-specific RNA-Seq studies of *P. falciparum* [59,60,61,62] and *T. gondii* (unpublished data, Chandra Ramakrishnan and Adrian Hehl and [63]). The full datasets show trends but are not entirely consistent for studies that encompass overlapping stages (Appendix A). Qualitative evidence for protein expression (yes/no) was obtained from deposited mass spectrometry (MS) peptide datasets at VEuPathDB.org [52]. The appearance of one or more unique peptides that map to a given tubulin for a life cycle form was taken as evidence for protein presence in this stage.

## 3. Results

### 3.1. Apicomplexans Display Simple α-, β-, and γ-Isotype Repertoires

Tubulins are highly conserved proteins with several characteristic motifs which permit family members to be identified as belonging to one of six groups within the tubulin superfamily. By using *Tetrahymena* and *Chlamydomonas* tubulins to query the genomes of apicomplexan parasites, we ascertained that individual species harbor one to four genes for α- and β-tubulins, whereas γ-tubulin is specified by a single gene in all cases (Table 1). Among the Apicomplexa, *P. falciparum* and *P. vivax* are agents of the most consequential human malarias. Both have two α-tubulin genes and a single β-tubulin gene, as do the important agricultural pathogens *T. annulata* and *B. microti*. These organisms are assigned to orders (Haemosporida and Piroplasmida) within the Aconoidasida class of apicomplexans (Figure 2). Species classified within the Conoidasida harbor genes for one to four α- and one to three β-tubulin isotypes. The presence of genes for multiple tubulin isotypes is noteworthy. Because apicomplexans are haploid, conserved α- or β-tubulin isotypes with similar expression patterns would permit functional redundancy, essentially making the organism “merodiploid” for this tubulin family. Conversely, if isotypes display distinct amino acid sequences and/or differences in expression, they likely function in biochemically distinct and/or stage-specific roles.

In the Aconoidasida, a single β-tubulin gene functions for all tubulin-requiring contexts. Although most coccidians have three β-tubulin isotypes, the amino acid sequences of these are nearly identical to each other, as illustrated by the alignment of the *T. gondii* isotypes (Appendix A). A genome-wide CRISPR screen performed on in vitro *Toxoplasma* tachyzoite cultures indicates that β3-tubulin is dispensable for growth in tachyzoites, whereas β1- and β2-tubulins have comparable negative fitness scores, indicating essential functions in tachyzoites [64]. The observation that *Toxoplasma* β3-tubulin expression is elevated during gametogenesis (see below) suggests that there may be a subtle feature of β3- that is critical for gametes. Notably, an essential sperm-specific isotype has been documented in a distinct biological context: testis-specific β-tubulin is required for the formation of the flagellar axoneme during *Drosophila* spermatogenesis [65].

In general, apicomplexan α-tubulins are much more diverse than β-tubulin isotypes. Aconoidasida species (*Plasmodium, Babesia, Theileria*) each harbor genes for two α-tubulin isotypes. Approximately 30 years ago, researchers recognized that *P. falciparum* α-tubulin II localized to the microgamete axoneme [66]. More recent studies reveal that α-tubulin II is co-expressed with α-tubulin I in other life cycle stages [67,68,69]. Isotypes I and II are 94% and 95% identical in *P. falciparum* and *P. vivax*, respectively. Alignments indicate that the H1-S2 (N) loop has the most differences between α- I and II. While this is more generally the region of greatest variation across all α-tubulins, it is also an area that is critical for lateral contacts between adjacent microtubule protofilaments. It is possible that, because flagellar axoneme microtubules are non-dynamic, the H1-S2 loop amino acids (QVVAGG) in α-tubulin II increase microtubule stability relative to α-tubulin I (KASRAN). In *T. annulata* and *B. microti*, one isotype harbors an insert of two or three amino acids in the H1-S2 loop relative to the other isotype. Inserted residues are not conserved between species, nor do they appear to correspond to the heterogeneity in the *Plasmodium* H1-S2 loops.

**Figure 2 microorganisms-11-00706-f002:**
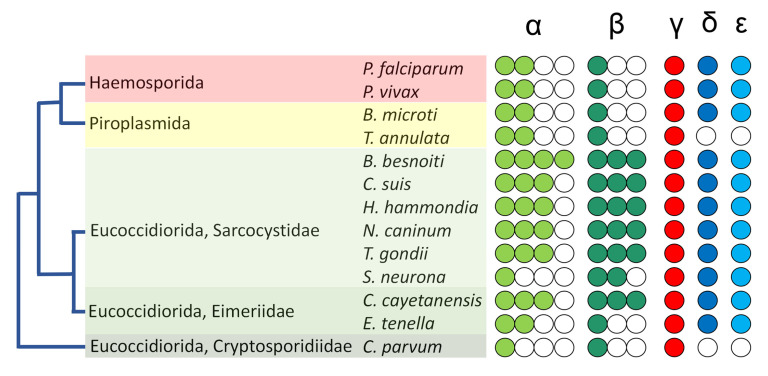
Tubulin superfamily members in apicomplexan species. *Plasmodium, Babesia and Theileria* species are classified in the Haemosporida and Piroplasmida, which are closely affiliated orders. The remaining apicomplexans analyzed in this study are classified as order Eucoccidiorida. Individual species have 1–4 α-tubulins and 1–3 β-tubulins (green). All have single genes for γ-tubulin (red), and most have single genes for δ- and ε-tubulin (blue). *Cryptosporidium* and *Theileria* lack genes for δ- and ε-tubulin, and their distinct positions in phylogenies suggest that these represent separate losses of δ- and ε-tubulin genes. Relationships between apicomplexan genera represent the basic organization of species from phylogenetic analyses, such as in the study by Kuo [68].

Members of the Conoidasida harbor genes for one to four α-tubulin isotypes. Protein alignments reveal that the three *Toxoplasma* α-tubulin isotypes are extremely dissimilar (Figure 3). The α1- isotype (TGME49_316400) is most like conventional α-tubulins. However, α2- (TGME49_231770) and α3- (TGME49_231400) contain unique insertions and substitutions, suggesting that they do not serve housekeeping roles. For example, α2-tubulin has a three amino acid insert (GGD) in the H1-S2 loop, which may increase stability during microtubule deformation. There are three alterations to α3-tubulin motifs that may dramatically influence its expression and function. First, α3-harbors an insert to the amino (N)-terminal tubulin regulatory sequence (MREI to MPREI), which is required to regulate tubulin abundance in many systems [70]. It should be noted that this insert changes the absolute numbering of subsequent residues, which is important for describing the second difference in α3-tubulin. Genetic and biochemical studies indicate that the highly conserved α-tubulin residues D251 and E254 are essential for the GTPase activating (GAP) domain function [71]. The GAP domain stimulates polymerization-dependent GTP hydrolysis in an adjacent β-tubulin heterodimer to permit microtubule disassembly. These positions are occupied by T252 (251) and H255 (254) in α3-tubulin, although there is a glutamic acid residue at position 254 (253; Figure 3, box 4). Loss and/or repositioning of these canonical residues suggests that α3-tubulin has a reduced or nonexistent capacity to stimulate GTP hydrolysis. Finally, α3-tubulin has an extraordinarily long carboxy (C)-terminal tail (~40 residues) not observed in other tubulins. These differences may imply that α2- and α3- influence microtubule function in unconventional ways, such as in non-dynamic and/or long-lived microtubules in bradyzoites, sporozoites or microgametes. Alternatively, they may indicate that α2- and α3-tubulins are in the process of becoming non-functional and dispensable isotypes.

When the *Neospora*, *Besnoitia*, *Hammondia*, and *Cystoisospora* α-tubulin homologs are compared to the *Toxoplasma* isotypes, isotype-specific features are conserved. *Hammondia* and *Besnoitia* α2-tubulin homologs HHA_231770 and BESB_009760 retain the H1-S2 loop insert, and the *Cystoisospora* homolog (CSUI_001393) has a larger insert of seven amino acids. Unfortunately, the *Neospora* sequence (NCLIV_031800) is incomplete and missing this region. The *Hammondia*, *Neospora*, *Besnoitia*, and *Cystoisospora* α3- homologs (HHA_231400, NCLIV_031660, BESB_009930, CSUI_005294) possess unusually long C-terminal tails, albeit with little shared sequence identity. All are missing the canonical GTPase activating residues D251 and E254 (replaced by T and Q/H). Although the N-terminal region of *Cystoisospora* α3- is missing, the N-terminal tubulin regulatory sequence insertion (MREI to MPREI) is present in the other three homologs. *Besnoitia* has a fourth α-tubulin (BESB_039600) that is most like α1-tubulin, although it harbors a substitution in the N-terminal tubulin autoregulatory sequence (MKEI rather than MREI/V). Of the two *E. tenella* α-tubulins, ETH2_1417400 is most like conventional α1-tubulins and ETH2_0609600 has similarities to α3-, such as an insertion to the N-terminal tubulin regulatory sequence (MPGEV) and the loss of the GTPase activating residues (replaced by C and S). The single α-tubulins in *S. neurona* and *C. parvum* are conventional (α1-like) sequences, with *Cryptosporidium* α-tubulin being the most divergent of the coccidian α1-tubulins.

The head to tail association of α-β heterodimers endows microtubules with structural and biochemical polarity (Figure 1). Microtubules may be anchored to microtubule organizing centers (MTOCs). In this context, the plus (fast polymerizing) end is distal to the MTOC, with β-tubulins outermost at the microtubule tip. The minus end is often embedded in a γ-tubulin containing complex, such that terminal α-tubulin subunits interact with γ-tubulin. All apicomplexans have single genes for γ-tubulin. The alignment of the *Toxoplasma*, *Cryptosporidium* and *Plasmodium* γ-tubulin sequences with γ-tubulin from *T. thermophila* show reasonably close conservation. In general, γ-tubulin alignments agree for the first 350–400 amino acids, with heterogeneity in the C-terminal sequences that suggest that the last 50–70 amino acids are missing from some predicted proteins (Table 1).

### 3.2. Not All Apicomplexans Have Genes for δ- and ε-Tubulins

Most surveyed apicomplexans harbor genes for apparent δ- and ε-tubulin homologs (Table 2). The exceptions are *C. parvum* and *T. annulata*, which lack genes for δ- and ε-tubulins. This absence was noted by Hodges and colleagues in a 2010 study of the evolutionary history of the centriole [22] and remains consistent in our reanalysis despite advancements in genome sequencing, assembly, and annotation for these genera. Species that lack δ- and ε-tubulin genes are members of two distinct apicomplexan lineages. The well-separated positions of *Theileria* (Piroplasmida) and *Cryptosporidium* (Eucoccidiorida) in phylogenetic analyses [72,73,74,75] are consistent with the conclusion that these genera independently lost δ- and ε-tubulin (Figure 2).

Meaningful comparisons of predicted δ- and ε-tubulin proteins from apicomplexans are limited by the lower level of homology among members of these families. However, it is notable that the tubulin signature motif, which is a highly conserved portion of the GTP-binding domain [2], deviates from the consensus or is missing altogether in some apicomplexan δ- and ε-tubulins (Figure 4). The tubulin signature appears to be entirely absent from the apparent *S. neurona* δ-tubulin homolog. *Plasmodium* δ-tubulin homologs have a non-canonical tubulin signature (AGGSGSG) rather than the expected [S/A/G]-G-G-T-G-[S/A]-G motif, and the *C. suis* signature has deteriorated to SEGGGSG. Surprisingly, the signature is preserved in *B. microti* δ-tubulin, despite studies that explicitly state that gametes are not flagellated. All apicomplexan ε-tubulins harbor a conserved tubulin motif, except for the apparent *S. neurona* homolog where, again, this sequence is undetectable. The low level of homology for *S. neurona* δ- and ε-tubulins and the apparent absence of a tubulin signature motif may indicate that these proteins are drifting into non-functional forms that no longer bind GTP.

### 3.3. Tubulin Expression Is Developmentally Regulated during the Parasite Life Cycle

To date, RNA-Seq and mass spectrometry (MS) studies have been carried out for several key *P. falciparum* and *T. gondii* stages. These data provide some insight into the expression of tubulin genes in zoite and gametocyte (pre-gamete) forms [59,60,61,62,63]. However, there are several caveats to these data. For example, although γ-tubulin is essential, peptides derived from it are not identified in all life cycle stages. This disparity likely represents a failure to detect peptides from less abundant proteins in stages that are technically more challenging to purify. Conversely, MS datasets for *Toxoplasma* tachyzoites and bradyzoites detect low levels of unique α2- and α3- peptides, while normalized RNA-Seq profiling does not detect significant transcripts for these tubulins. Lastly, while the *Toxoplasma* data are represented as normalized counts and were generated by the same research lab using comparable workflows, the *Plasmodium* data are from different labs and were not similarly processed by the original investigators. Despite these shortcomings, the existing highly normalized data extracted from VEuPathDB suggests that the transcription of δ- and ε-tubulin genes in *Plasmodium* is specific to microgametocytes, whereas the synthesis of δ- and ε-tubulin proteins in *Toxoplasma* may be regulated post-transcriptionally.

Late trophozoite and oocyst stage *P. falciparum* samples have comparable levels of β-, α-I, and α-II, and slightly lower levels for γ-tubulin transcripts (Figure 5). Peptides for α-I, α-II, and β- are present in red cell stage parasites (e.g., trophozoites) and in oocysts. As expected, given the absence of flagella in vegetative forms, δ- and ε-tubulin transcripts and peptides are absent or negligible in red cell stages and oocysts. While macrogametocytes have low levels of δ- and ε-tubulin transcripts, levels of α-II, δ-, and ε-tubulin are over a magnitude higher in microgametocytes. Moreover, δ- and ε-tubulin peptides are specifically detected in microgametocyte samples. These data are consistent with the conclusion that the *Plasmodium* expression of δ- and ε-tubulins is microgamete-specific.

Tachyzoites, bradyzoites and oocysts represent distinct *T. gondii* asexual forms obtained generally as highly enriched, defined stages. On the other hand, the EES5 sample is composed of a mixture of merozoites, microgametocytes, macrogametocytes, macrogametes, and microgametes at the late stage of enteroepithelial development in the cat intestine. Transcripts for the three β-tubulins and for γ-tubulin are found in all four samples (Figure 5). Transcripts for the unusual α2- and α3-tubulins are negligible or absent in tachyzoites, and α2- mRNA is also absent from bradyzoites. Surprisingly, δ- and ε-tubulin transcripts are found in all four samples, although there is no peptide evidence for protein expression in tachyzoite, bradyzoite and oocyte samples. Currently, it is not possible to obtain enriched gametocyte samples for MS studies to date.

## 4. Discussion

Tubulins are an ancient superfamily of GTP-binding proteins that emerged prior to the divergence of extant eukaryotes [5,20,76]. The highest sequence identity is found in the α-, β-, and γ-tubulin families, which are especially constrained by requisite interactions with GTP, other tubulins, and other proteins. In all eukaryotes, γ-tubulin nucleates microtubules that assemble from α-β-tubulin heterodimers. Their ubiquitous presence and sequence conservation makes α-, β-, and γ-tubulins valuable for molecular systematics of eukaryotes. The remaining three families, δ-, ε-, and ζ-tubulins, are components of centrioles and basal bodies, ancient organelles found in many eukaryotic groups, which suggests that they also emerged at around the time of the last eukaryotic common ancestor. ZED tubulins have been lost from diverse eukaryotic lineages [6,7,18]. These organisms illustrate that there are many routes to survive the loss or reduction of centriole and basal body functions.

Centrioles and basal bodies are found in lower plants, chytrid fungi, animals, and many protozoa. They typically consist of a cylinder of nine triplet microtubules (Figure 1) but may also be comprised of nine doublet or singlet microtubules in rarer circumstances. The construction of a transition zone converts a centriole into a basal body, which has the capacity to template and anchor an axoneme at the plasma membrane. In some eukaryotes, duplicated centrioles are located at spindle poles and serve to organize signaling molecules during chromosome segregation and cytokinesis. Separated centrioles are subsequently converted into basal bodies to construct flagella in daughter cells (Figure 6). However, early branching eukaryotes employ a closed mitosis, which occurs without nuclear envelope breakdown. For many of these (e.g., *Tetrahymena*), spindle microtubules emanate from acentriolar plaques/spindle pole bodies (SPB) embedded in the nuclear envelope and basal bodies are constructed exclusively for axoneme functions [77,78]. Other protozoa (e.g., *Toxoplasma*) have both SPBs (centrocones) and associated centrioles [79].

Apicomplexan parasites are located within the Alveolata, a large and diverse group of eukaryotic microorganisms that also includes ciliates and dinoflagellates [80,81,82]. Many alveolates construct canonical 9 × 2 + 2 axonemes to form motile cilia or flagella [83]. In fact, the ciliates *Tetrahymena* and *Paramecium* are well-established model systems for the study of ciliary function [78,84]. The recent expansion of sequenced apicomplexan genomes provides an opportunity to analyze members of the tubulin superfamily relative to those represented in ciliates. A previous study evaluated several apicomplexans, along with *Tetrahymena* and *Paramecium*, in the context of reconstructing the evolutionary history of the centriole [22]. That work demonstrated that *Toxoplasma* and *Plasmodium* have fewer homologs for core (ancestral) centriole components relative to *Tetrahymena* and *Paramecium*. While ciliates such as *Tetrahymena* generally occupy an extracellular, predatory niche that requires persistent axoneme-based motility, apicomplexans are obligate intracellular parasites. The asexual (zoite) stages of apicomplexans are not flagellated; these forms may use an unusual actin-based gliding motility to move and to invade host cells [43]. Therefore, the function of flagella in apicomplexans is, at most, reduced to a role in microgamete fertilization during the transient sexual cycle. Some, but not all apicomplexans harbor genes for δ- and ε-tubulins, and some but not all apicomplexan microgametes construct centrioles, basal bodies and flagellar axonemes (Figure 6).

Comparative genome analysis and ultrastructural studies indicate that δ- and ε-tubulin genes have been separately lost from many eukaryotic lineages, including the genera of model organisms *Drosophila*, *Caenorhabditis*, and *Saccharomyces* [5,19,20]. In each case, the consequences of this loss are different. *Drosophila* centrioles lack subdistal appendages, structural elements that typically distinguish the mature (mother) centriole [21,85,86,87]. These nine triplet-containing structures are converted into basal bodies that template axonemes, including for motile flagella in sperm cells. *Caenorhabditis* centrioles consist of nine singlet microtubules, and its non-flagellated sperm move by a distinct crawling mechanism [23,88,89]. *C. elegans* constructs non-motile (9 × 2 + 0) sensory cilia in some neurons that may be highly branched and originate in basal bodies consisting of doublet microtubules. Lastly, *Saccharomyces* and many other fungi do not form centrioles, basal bodies, or flagella. The presence of ε- and ζ-tubulin genes, basal bodies, and motile flagella in early branching chytrids indicates that both tubulins and structures were lost from other fungal lineages [5,18,90]. These examples of δ- and ε-tubulin loss and centriole, basal body and axoneme adaptations have apparent parallels in apicomplexan organisms.

As apicomplexans transitioned to an intracellular lifestyle, their need for flagellar motility was reduced to a stage-specific role in microgametes during the fertilization of macrogametes [91]. This shift reduced the pressure for the maintenance of centriole, basal body and axoneme structures, which led to the simplification or loss of these structures. For example, asexual stage *Toxoplasma* constructs atypical singlet centrioles, whereas asexual stage *Plasmodium* lacks these altogether [41,79,92,93]. Therefore, while *Plasmodium* microgametes must undergo de novo basal body assembly (a process that also occurs in *Naegleria* and *Marsilea* [94,95]), *Toxoplasma* may specifically synthesize δ- and ε-tubulin proteins during gametogenesis to convert singlet centrioles to triplet basal bodies [50,96]. δ- and ε-tubulins in individual apicomplexan species may be altered, vestigial, or absent.

The fullest picture of the relationship between apicomplexan δ- and ε-tubulins and tubulin-based structures is achieved by integrating ultrastructural observations of specific life cycle forms with analysis of parasite genomes. Unfortunately, only a few electron micrographs capture cross-sections through apicomplexan microgamete basal bodies, which are often described as electron-dense [92,97,98,99,100]. In *Plasmodium*, images show singlet microtubules which may represent the mature basal body form or reflect an intermediate stage of assembly. Similarly, a basal body from a microgamete of the coccidian *S. suihominis* shows a combination of doublet and triplet microtubules. The apparent *Sarcocystis* δ- and ε-tubulin homologs lack a discernible tubulin signature, and the *Plasmodium* δ-tubulin signature is non-canonical (Figure 4). Mutations that compromise the function of δ- or ε-tubulin in *Chlamydomonas* reduce basal body triplets to doublets or singlets [101,102,103]. Within the Piroplasmida, both *Theileria* and *Babesia* microgametes lack a 9 × 2 + 2 motile axoneme structure found in many eukaryotes, including *Plasmodium* [97,104]. *T. annulata* microgametes are described as spindle-shaped, with a pointed apex and flagellar-like protrusions containing two to six microtubules [105]. Remarkably, although the *B. microti* genome harbors δ- and ε-tubulin genes, its gametes lack axonemes and basal bodies [106,107,108]. Indeed, Weber and Friedhoff explicitly state that the “so-called strahlen (spiky rays) are cytoplasmic processes supported by a bundle of long, parallel microtubules that are distributed randomly and not configurated to a 9 + 2 or another pattern” [108]. Lastly, in the case of *Cryptosporidium*, the loss of δ- and ε-tubulin genes is in line with an overall reduction in genome complexity, including the loss of centriole components [22,29]. Super-resolution (fluorescence) microscopy of *Cryptosporidium* microgametes indicates that they lack flagella but possess sets of microtubules that run along either side of the length of the elongated nucleus [109]. As with *Theileria* and *Babesia* microgametes, these microtubule bundles are likely relics of the flagella observed in *Plasmodium* and *Toxoplasma* microgametes.

Previous analysis of committed male and female *Plasmodium* gametocytes reveals that the synthesis of flagellar components is restricted to this stage [110,111]. Microgametocytes specifically synthesize δ- and ε- tubulin transcripts and proteins (Figure 5). Several lines of indirect evidence suggest that *Toxoplasma* may specifically synthesize δ- and ε-tubulin proteins in microgametes. *Toxoplasma* tachyzoites construct singlet centrioles with structural similarity to δ- and ε-tubulin-deficient *C. elegans* centrioles [79]. MS surveys do not detect δ- or ε-tubulin in tachyzoites or other asexual stages. Lastly, there is ultrastructural evidence for triplet microtubule basal bodies in a coccidian (*S. suihominis*) microgamete. To date, it has not been possible to culture or isolate pure committed gametocytes, macrogametes, or microgametes from *Toxoplasma* for RNA-Seq or MS analysis. Felid enteroepithelial samples contain merozoites, gametocytes, macrogametes, and microgametes [112]. Although this mixed population may obscure differences between macrogametes and microgametes, the globally similar transcript levels in this and other samples (tachyzoites, bradyzoites and unsporulated oocysts) suggest that the synthesis of δ- and ε-tubulin proteins is regulated at a post-transcriptional level. Once it is possible to obtain pure samples of these forms, these hypotheses can be tested.

Because fertilization is essential for the transmission of some apicomplexans, the disruption of basal body and axoneme assembly has obvious relevance for reducing the incidence of disease. SAS-6 is a conserved and essential component of centrioles and basal bodies that establishes the nine-fold symmetry required for both duplication and function. Like δ- and ε-tubulin, its expression is microgamete-specific in *Plasmodium*, which also supports the interpretation that basal bodies are assembled de novo at this stage [49]. Targeted gene knockouts for SAS-6 and PF16 in *Plasmodium* have shown that basal body and axoneme components are important for the transmission of infection [49,113]. The disproportionately large impact of malaria on human morbidity and mortality means that a future method to compromise *Plasmodium* microgamete flagellar assembly or function could be transformative for human health. However, the specific set of basal body components, the existence of flagellated microgametes, and the importance of fertilization for transmission vary between individual apicomplexan lineages. Therefore, strategies to target microgamete flagella would not be universally applicable for all apicomplexan pathogens.

The individual evolutionary trajectories of apicomplexan parasites have led them to occupy distinct host cell and tissue niches, and consequently to have different therapeutic vulnerabilities. Most standard-of-care treatments for apicomplexan infections have limited utility for the inhibition of other apicomplexans. For example, erythrocytic stage *Plasmodium* parasites degrade hemoglobin as a source of nutrients, and many anti-malarial drugs interfere with this process [114]. These drug targets are non-essential for apicomplexans that infect other host cell types. Similarly, nitazoxanide, which interferes with electron transfer during anaerobic metabolism, is used to treat *Cryptosporidium* and *Cyclospora* infections, which remain in the relatively hypoxic environment of the intestine [115]. Because *Toxoplasma* disseminates from the intestine to tissue sites with higher oxygenation, it is less reliant on anaerobic metabolism and nitazoxanide is ineffective. Daraprim, the standard-of-care drug for toxoplasmosis, inhibits folate metabolism [116]. *Cyclospora* can also be treated by inhibiting folate metabolism, but *Cryptosporidium* has streamlined metabolic pathways, including an unexpected ability to salvage purines from the host [117]. In contrast, microtubules represent a conserved broad spectrum drug target shared by all apicomplexan pathogens.

Small molecules that bind to tubulin heterodimers alter the dynamic properties of microtubules and are used in clinical settings to treat cancer, inflammatory disorders, and helminth (worm) parasite infection. Typically, microtubule targeting agents disrupt dynamic spindle microtubules to impair mitosis. Phylogenetic analysis indicates that apicomplexan tubulins are most like tubulins from ciliates and green algae [118,119,120]. Not surprisingly, the phylogenetic relationships of α- and β-tubulins align with sensitivity to dinitroanilines, small molecules that inhibit microtubule assembly. Dinitroanilines prevent mitosis in protozoan parasites, green algae, and land plants, but are inactive in vertebrates or fungi [50,51]. Other tubulin-targeting compounds (MMV676477 and parabulin) also selectively inhibit apicomplexans and other protozoa [121,122]. Genes for distinct α- and β-tubulin isotypes may influence drug resistance. For example, aggressive and treatment refractory human cancers increase the expression of less sensitive isotypes to reduce the efficacy of tubulin targeting drugs [123]. This mechanism is less likely to arise in apicomplexans, which have simpler tubulin gene organization. However, drug resistance may arise by missense mutations that alter tubulin biochemistry, as observed for *Toxoplasma* oryzalin resistance which is conferred by α1-tubulin substitutions [124]. However, since apicomplexans are haploid, the co-expression of conserved isotypes may suppress the ability of missense mutations to confer resistance. Microtubules composed of a mixture of sensitive and resistant proteins remain somewhat susceptible to disruption. We hypothesize that drugs that bind to co-expressed tubulin subunits (e.g., *Toxoplasma* β-tubulins and possibly *Plasmodium* α-tubulins) will suppress the development of drug resistant strains.

Available genomes, transcript profiling, and morphological studies provide critical information on the organization, properties, and expression of tubulin superfamily members in apicomplexans. While the present data can be incorporated into a logical narrative, other details remain unanswered until researchers are able to collect robust and specific datasets for additional (currently inaccessible) life cycle stages. Nonetheless, existing evidence validates tubulin and microtubules as critical targets for anti-parasitic therapies and transmission-blocking strategies to control apicomplexan pathogens.

## Figures and Tables

**Figure 1 microorganisms-11-00706-f001:**
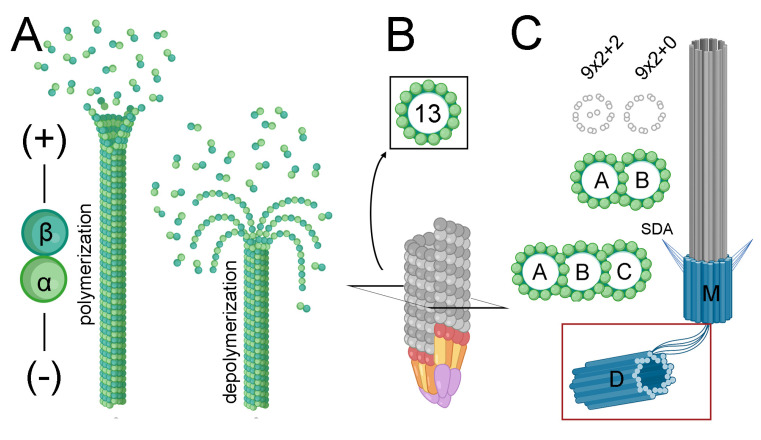
Contribution of tubulin superfamily members to microtubule structures. (**A**) Microtubules are assembled by the reversible assembly of α-β tubulin heterodimer subunits (green). Tubulin heterodimers stack head to tail and these protofilaments associate laterally to form microtubules, which close at a seam located at a lateral junction of α- and β- interfaces. During depolymerization, protofilaments separate from each other and curl away from the microtubule tip prior to the dissociation of individual heterodimers. Microtubules are polarized, with β-tubulin subunits exposed at the fast-polymerizing (+) end and α-tubulin located at the tip of the (-) end. (**B**) Microtubules are typically templated by nucleation from the γ-tubulin ring complex (γ-TuRC), which consists of γ-tubulin (red) and associated proteins (yellow, orange, purple). This lock washer-like structure nucleates 13-protofilament microtubules by interaction of γ-tubulin with α-tubulin. (**C**) Axonemes extend from basal bodies (blue), which consist of a centriole and associated transition zone. Centrioles are formed by semi-conservative replication: the older mother (M) centriole contains subdistal appendages (SDA) that are absent from the younger daughter (D) centriole. While centrioles remain connected by proximal end filaments in non-motile sensory (9 × 2 + 0) cilia, they are disconnected in motile (9 × 2 + 2) flagella. Moreover, although metazoan sperm build a single flagellum, unicellular organisms including *Plasmodium* microgametes and *Chlamydomonas* have two flagella, with axonemes appended to both mother and daughter centrioles. Most basal bodies are constructed from 9 triplet microtubule blades (9 × 3) and require the expression of ε- and δ- or ζ- tubulins to assemble correctly. The motile flagellar axoneme consists of a central pair of singlet microtubules with nine surrounding doublet microtubules. Immotile axonemes in sensory cilia lack the central pair. Doublet and triplet microtubules consist of a conventional 13 protofilament microtubule (the A tubule) which shares a partial wall with an 11 protofilament “B” tubule. Triplet microtubules have a second 11 protofilament “C” tubule appended to the B tubule. Figure generated with images from Biorender.

**Figure 3 microorganisms-11-00706-f003:**
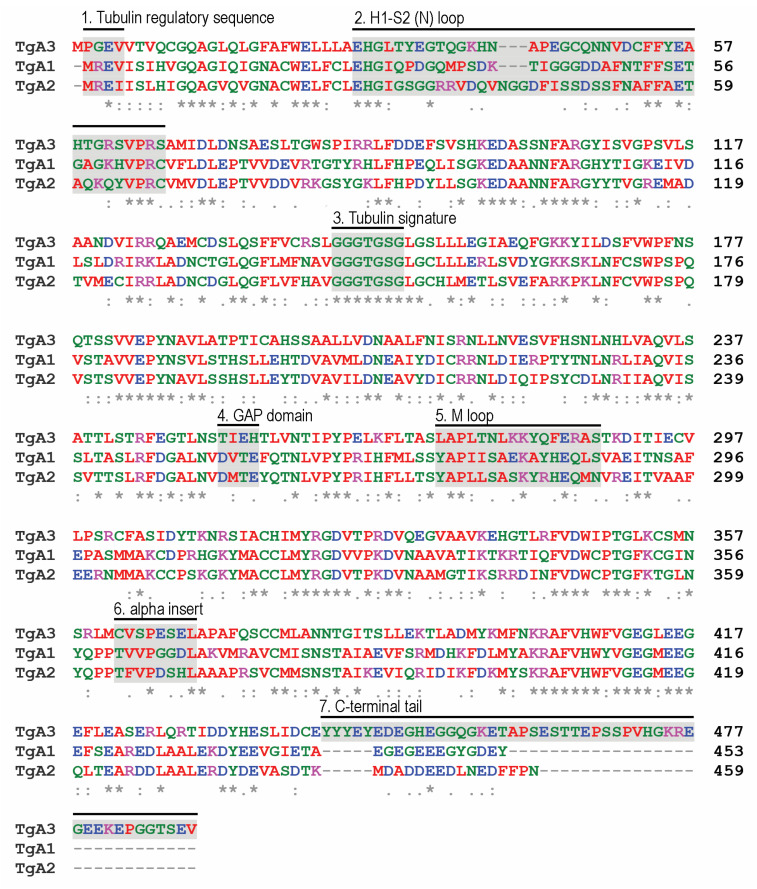
Alignment and features of the three *Toxoplasma* α-tubulin isotypes. The numbered boxes indicate motifs found in α-tubulins. While the tubulin signature is conserved in all three isotypes (3), other features of α2- and α3- are unusual. The M and H1-S2 loops (boxes 2 and 5) coordinate interactions between protofilaments and are not conserved with α1-tubulin. All α-tubulins have an 8 amino acid insert, which is thought to stabilize M loop interactions between protofilaments. Its sequence (6) is not conserved between isotypes. The GTPase activating domain (GAP, box 4) requires D251 and E254 to activate polymerization-dependent hydrolysis, which is essential for microtubule disassembly. The absence of D251 and the shift of E254 suggest that the α3-isotype has slowed, or non-existent GAP activity which may stabilize α3-containing polymers. A3-tubulin has an insertion to the tubulin regulatory sequence (MREI, box 1), which controls cytoplasmic levels of tubulin heterodimers, suggesting it may not be responsive to the same regulation. In addition, α3-tubulin has an extraordinarily long C-terminal tail (48 amino acids, box 7) not observed in other characterized tubulins. The notation “*” indicates a completely conserved residue, “:” indicates conservation between groups of strongly similar properties, and “.” denotes conservation between groups of weakly similar properties.

**Figure 4 microorganisms-11-00706-f004:**
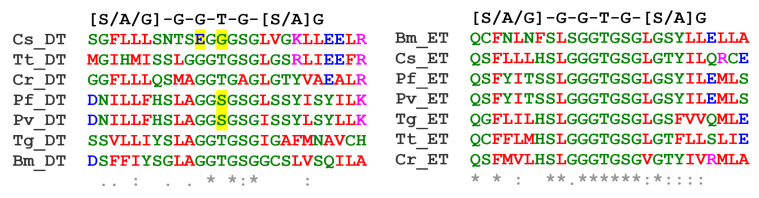
Alignment of the tubulin signature motif in δ- and ε-tubulins from the unicellular green alga *C. reinhardtii* (Cr), the ciliate *T. thermophila* (Cr), and the apicomplexans *B. microti* (Bm)*, C. suis* (Cs)*, T. gondii* (Tg)*, P. falciparum* (Pf), and *P. vivax* (Pv). The ε-tubulin (ET) motif is intact (**right**), but the δ-tubulin (DT) signature (**left**) has diverged in *Plasmodium* and *Cystoisospora* which may reduce GTP binding affinity. Although we predict that δ- and ε-tubulins are relict proteins in *Babesia*, they retain a consensus tubulin signature. The notation “*” indicates a completely conserved residue, “:” indicates conservation between groups of strongly similar properties, and “.” denotes conservation between groups of weakly similar properties. The highlighted amino acids in δ-tubulins indicate deviations from the tubulin signature.

**Figure 5 microorganisms-11-00706-f005:**
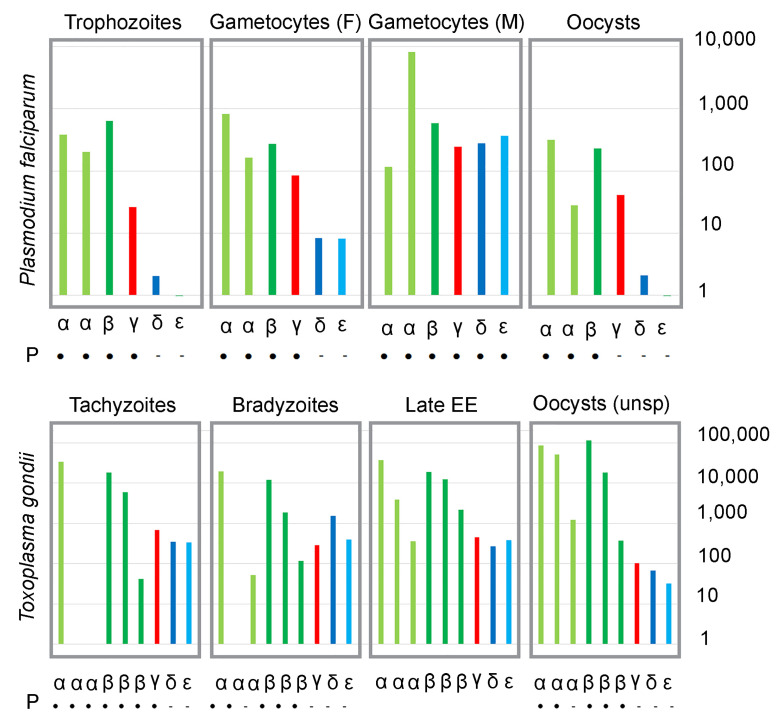
Expression of tubulin isotypes and family members in *P. falciparum* and *T. gondii*. (**Top**) Transcript abundance in late trophozoite, macrogametocyte, microgametocyte, and oocyst stages of *P. falciparum*. The presence (•) or absence (-) of unique peptides (P) for individual tubulins is indicated below the transcript data. The apparent absence of peptides for γ-tubulin in oocysts is likely due to sensitivity rather than lack of expression. Note the increased mRNA abundance of αII-, δ-, and ε-tubulins in microgametocytes relative to other stages. (**Bottom**) Transcript abundance in tachyzoites, bradyzoites, late enteroepithelial (EE) stages and unsporulated (unsp) oocysts from *T. gondii*. The presence or absence of unique peptides (P) for individual tubulins is indicated below the transcript data. At present, no MS dataset is comparable to the late EE stage sample. The apparent absence of peptides for γ-tubulin in bradyzoites and oocysts is likely to be due to detection issues rather than lack of expression. Note that there are comparable mRNA levels for δ- and ε-tubulins in all samples, indicating that transcription is not likely to be developmentally regulated.

**Figure 6 microorganisms-11-00706-f006:**
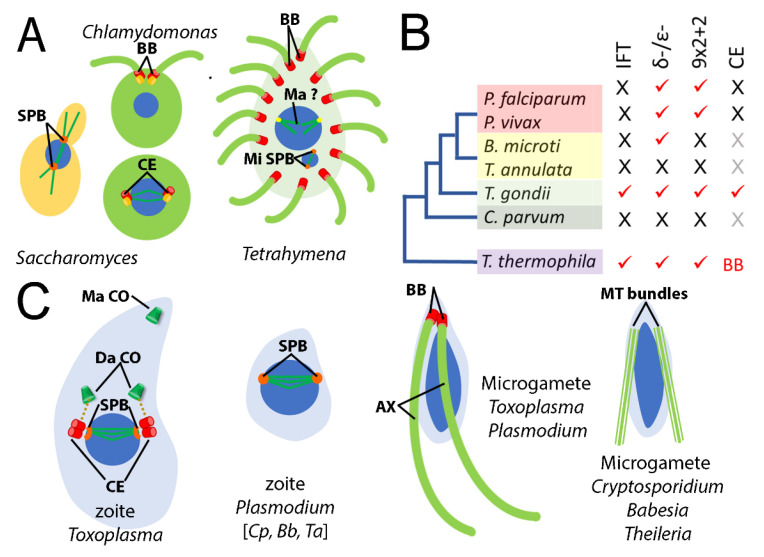
Organization of spindle poles, centrioles, basal bodies, and flagella in unicellular eukaryotes. (**A**) Distinct microtubule organization in the unicellular model organisms *S. cerevisiae*, *C. reinhardtii*, and *T. thermophila*. *Saccharomyces* lacks δ- and ε-tubulin, basal bodies, and flagella. Its spindle microtubules are organized by SPBs. *Chlamydomonas* harbors δ- and ε-tubulin genes and constructs two 9 × 2 + 2 flagella which emanate from basal bodies (BB). During cell division, flagella are resorbed and basal bodies become centrioles (CE), located at the poles of the spindle. *Tetrahymena* possesses two nuclei and is covered with cilia which originate in arrays of basal bodies. The germline (micronucleus) spindle emerges from SPB-like structures (Mi SPB), whereas organization of the somatic (macronuclear, Ma) spindle is poorly understood. (**B**) Variations in tubulins and tubulin structures in apicomplexan parasites and the related alveolate *Tetrahymena*. Intraflagellar transport (IFT) components, δ- and ε-tubulins, a 9 × 2 + 2 flagellar axoneme, and spindle associated centrioles are cataloged: red check marks indicate presence, while crosses indicate demonstrated (black) and inferred (gray) absence of characteristics. (**C**) Spindles, centrioles, centrocone (SPB) and axonemes in apicomplexans. *Toxoplasma* asexual stage zoites replicate by forming two internal daughter parasite buds. Spindle microtubules enter the nucleus through centrocones and are associated with centrioles at their poles. Centrioles are linked to emergent sets of daughter apical organelles, including the conoid (Da CO). *Plasmodium* zoites lack centrioles; their spindle microtubules emerge from centrocones. Centrioles have not been reported in ultrastructural studies of *Cryptosporidium*, *Babesia*, and *Theileria* zoites; therefore, their spindle poles are likely to be organized by centrocone plaques. Both *Toxoplasma* and *Plasmodium* microgametes are flagellated, with 9 × 2 + 2 axonemes. *Cryptosporidium*, *Babesia*, and *Theileria* microgametes are not flagellated, but have bundles of microtubules flanking the nucleus which likely represent vestigial axonemes.

**Table 1 microorganisms-11-00706-t001:** Universal tubulins in apicomplexan species.

Species	α-Tubulin	β-Tubulin	γ-Tubulin
*Babesia microti*	BmR1_04g09455 Ch IV *BMR1_02g00915* ^1^ Ch II	*BMR1_01G01045* ^7^ Ch I	*BmR1_04g07605* ^10^ Ch IV
*Besnoitia besnoitia*	BESB_078340 Ch VII BESB_039600 ^2^ Ch II BESB_009760 Ch I BESB_009930 Ch I	BESB_049820 Ch III BESB_055160 Ch IV BESB_080330 Ch VII	*BESB_059730* ^11^ Ch V
*Cryptosporidium parvum*	cgd4_2860 Ch 4	cgd6_4760 Ch 6	*cgd7_1980* ^12^ Ch 7
*Cyclospora cayetanensis*	cyc_07446 NA *cyc_01940* ^3^ NA	cyc_02710 NA cyc_00127 NA	*cyc_08632* ^13^ NA
*Cystoisospora suis*	CSUI_008696 NA CSUI_001393 NA *CSUI_005294* ^4^ NA	CSUI_006267 NA CSUI_009771 NA CSUI_006169 NA	CSUI_002962 NA
*Eimeria tenella*	ETH2_1417400 Ch 14 *ETH2_0609600* ^5^ Ch 6	ETH2_0401900 Ch 4	*ETH2_1576800* ^14^ Ch 15
*Hammondia hammondia*	HHA_316400 NA HHA_231770 NA HHA_231400 NA	HHA_266960 NA HHA_221620 NA HHA_212240 NA	HHA_226870 NA
*Neospora caninum*	NCLIV_058890 Ch XI *NCLIV_031800* ^6^ Ch VIII NCLIV_031660 Ch VIII	NCLIV_039100 Ch IX NCLIV_005150 Ch II NCLIV_049140 Ch X	NCLIV_046130 Ch X
*Plasmodium falciparum*	PF3D7_0903700 Ch 9 PF3D7_0422300 Ch 4	PF3D7_1008700 Ch 10	PF3D7_0803700 Ch 8
*Plasmodium vivax*	PVP01_0702100 Ch 7 PVP01_0530800 Ch 5	PVP01_0808400 Ch 8	PVP01_0116500 Ch 1
*Sarcocystis neurona*	SN3_01900485 NA	SN3_00103320 NA SN3_02800250 ^8^ NA	*SN3_01300465* ^15^ NA
*Theileria annulata*	TA08335 Ch 4 TA21240 Ch 1	*TA13315* ^9^ Ch 2	*TA03470* ^16^ Ch 3
*Toxoplasma gondii*	TGME49_316400 Ch XI TGME49_231770 Ch VIII TGME49_231400 Ch VIII	TGME49_266960 Ch IX TGME49_221620 Ch II TGME49_212240 Ch X	TGME49_226870 Ch X

Italicized accession numbers appear incomplete or inaccurately annotated. Ch: chromosome location; NA: chromosome location not assigned. **^1^** BMR1_02g00915 may be missing ~15 amino acids from the C-terminal tail. **^2^** BESB_039600 is a second α1-like tubulin. **^3^** cyc_01940 is fusion of a partial α-tubulin gene with Sec1 gene, partial sequence. **^4^** CSUI_005294 lacks an N-terminal region and has excess amino acids at the C-terminal region. **^5^** ETH2_0609600 appears to have two internal inserts (72, 42) and is missing a C-terminal domain. **^6^** NCLIV_031800 is missing 69 N-terminal amino acids. **^7^** BMR1_01G01045 is likely missing 6-10 carboxy terminal amino acids. **^8^** SN3_02800250 has an atypical insert of 88 residues beginning at position 33 (may be inaccurate). **^9^** TA13315 is likely missing 6-10 carboxy terminal amino acids. **^10^** BmR1_04g07605 is likely missing 4 N-terminal and 10 C-terminal amino acids, also internal gaps. **^11^** BESB_059730 may have ~20 excess amino acids at the C-terminal region. **^12^** cgd7_1980 likely has ~20 excess amino acids at the C-terminal region. **^13^** cyc_08632 has an atypical 25 amino acid insert at position 329. **^14^** ETH2_1576800 is missing ~8 amino acids at position 382. **^15^** SN3_01300465 appears to have artifactual inserts: 101-186; 301-318; 417-449. **^16^** TA03470 has an atypical 35 amino acid insert at position 240.

**Table 2 microorganisms-11-00706-t002:** δ- and ε-tubulins in apicomplexan species.

Species	δ-Tubulin	ε-Tubulin
*Babesia microti*	BMR1_02g03305 Ch II	BMR1_03g04130 Ch III
*Besnoitia besnoitia*	BESB_022320 Ch XII	BESB_029970 Ch XIII
*Cryptosporidium parvum*	*Not identified*	*Not identified*
*Cyclospora cayetanensis*	cyc_04504 ^1^ NA	cyc_05034 NA
*Cystoisospora suis*	CSUI_009859 ^2^ NA	CSUI_001319 NA
*Eimeria tenella*	ETH2_1040000 Ch 10	ETH2_0530700 Ch 5
*Hammondia hammondia*	HHA_207600 NA	HHA_275870 NA
*Neospora caninum*	NCLIV_002570 ^3^ Ch Ib	NCLIV_007070 Ch III
*Plasmodium falciparum*	PF3D7_0933800 ^4^ Ch 9	PF3D7_1475700 Ch 14
*Plasmodium vivax*	PVP01_0732500 ^5^ Ch 7	PVP01_1270600 Ch 12
*Sarcocystis neurona*	SN3_00102395 ^6^ NA	SN3_01100650 ^7^ NA
*Theileria annulata*	*Not identified*	*Not identified*
*Toxoplasma gondii*	TGME49_207600 Ch Ib	TGME49_275870 ^8^ Ch III

Ch: chromosome location; NA: chromosome location not assigned. **^1^** cyc_04504 annotated as “tubulin gamma related” but is δ-tubulin. **^2^** CSUI_009859 has degenerate tubulin signature (ASEGGGSG). **^3^** NCLIV_002570 the 19 residue N-terminal extension is likely an artefact. **^4^** PF3D7_0933800 has degenerate tubulin signature (AGGSGSG). **^5^** PVP01_0732500 has a degenerate tubulin signature (AGGSGSG). **^6^** SN3_00102395 lacks a detectable tubulin signature. **^7^** SN3_01100650 lacks a detectable tubulin signature. **^8^** TGME49_275970 was reannotated to include an N-terminal region.

## Data Availability

All data supporting reported results can be found at VEupathDB.org or in supporting data for the papers cited in the methods.

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
