# Peer review of "The Tubulin Superfamily in Apicomplexan Parasites"

_microorganisms, 2023, doi:10.3390/microorganisms11030706_

Round 1

Reviewer 1 Report

The paper Morrissette et al. is devoted to the very important issue of correlating the expression of various genes of the tubulin family and the functional features of centrosomes, cilia and flagella in various organisms. When reading the article, there is no doubt about the competence of the authors in matters of parasitology, biochemistry and molecular biology of the studied objects. Unfortunately, in the field of morphology, the article has significant shortcomings that need to be corrected before the article can be published in the journal Microorganisms. My special articles were devoted to correcting mistakes and inaccuracies made by the authors in the article, so it makes no sense to keep the reviewer anonymous in this case, since I recommend the authors to read my works.

Figа 1С. Line 67

I want to draw your attention to the fact that the direction of twisting (tilt) of triplets in centrioles is not arbitrary, but strictly determined. In your drawing, the child centriole is oriented toward the reader with its proximal end. In this case, the triplets should be shown twisted counterclockwise (in your drawing, it's the other way around). There is a special article on this issue.

Clockwise or anticlockwise? Turning the centriole triplets in the right direction!

Uzbekov R, Prigent C. FEBS Lett. 2007 Apr 3;581(7):1251-4. doi: 10.1016/j.febslet.2007.02.069. Epub 2007 Mar 6. PMID: 17368628 Free article. Review.

In addition, you write that you have depicted a mobile eyelash in Figure 1C: «Motile
flagellar axonemes extend from basal bodies (blue) ...»

and in fact the basal body of the motile cilium is not associated with the second (daughter) centriole. With the exception of the two central microtubules of the axoneme, your drawing shows the complex of the basal body and the primary (non-motile, sensory) cilium. See for more details:

Principal Postulates of Centrosomal Biology. Version 2020.

Uzbekov RE, Avidor-Reiss T. Cells. 2020 Sep 24;9(10):2156. doi: 10.3390/cells9102156. PMID: 32987651 Free PMC article. Review.

Thus, if you want to show the primary cilium in this picture, then you need to remove the central microtubules of the axoneme, and if you want to show the mobile cilium, then you need to remove the second centriole. Of course, the cilium-basal body complex is not limited to this simplified scheme. There are additional structures on the basal body (rootlets, foot...), but authors can give a more complete or simplified scheme. Main, so that there are no errors.

Figure 1C gives an idea of the structure of the cilium-basal body complex in the cells of multicellular animals (more precisely, even in vertebrates). In various protozoans, which are the main objects of this study, the structure of this complex has its own characteristics, which the authors partially demonstrate in Figure 6. For example, two flagella in Chlamydomonas ...

This discrepancy may cause readers to misunderstand some of the provisions expressed in the article. It is necessary to give more detailed schemes of the cilium-basal body complex in precisely those organisms to which the article is devoted.

Line 86-87

Next, you write about appendages, in the diagram they are shown, but not indicated in any way.

« Loss of δ- and ε-tubulins is associated with an inability to form appendage-containing basal bodies, or to build basal bodies and motile flagella [20-23].

Line 284

“Most microtubules are anchored to microtubule organizing centers (MTOCs).

This is too categorical statement, because it is not true for most cell types. Most microtubules in the cytoplasm of cells are "free". They either broke away from the microtubule organization centers or were polymerized independently of them. Here it is necessary to indicate specifically those types of cells for which the authors are sure that their statement is true.

Fig. 6 Lines 405 and 411 and 420 and 478 and 484

425,

It is better to use the full formula reflecting the total number of microtubules in the axoneme  (9Ñ…2+2).  

Line 444-447

«Although Drosophila sperm have motile flagella, their basal bodies lack subdistal appendages, structural extensions that typically stabilize

the mature (mother) centriole/basal body and coordinate oriented connection to the plasma membrane [21, 79-81].»

In the basal bodies of spermatozoa of any organism, as far as I know, there are no subdistal appendages ("foot", "arm"). The function of subdistal appendages is completely different in centrosomes. You probably mean here the analog of subdistal appendages in the basal bodies ("foot" in another terminology "arm" ). In the ciliary epithelium, they are indeed oriented identically for all cilia in the cell.

About subdistal appendages there is special paper:

Who are you, subdistal appendages of centriole?

Uzbekov R, Alieva I. Open Biol. 2018 Jul;8(7):180062. doi: 10.1098/rsob.180062. PMID: 30045886 Free PMC article. Review.

Line 449

t is better to use the full formula reflecting the total number of microtubules in the axoneme (9x2+0)

The text of the paper a lot of word hyphenation from line to line. I would recommend authors to avoid this by using the justifier function.

Author Response

Please see attached highlighted file of MS

Reviewer 1: The paper Morrissette et al. is devoted to the very important issue of correlating the expression of various genes of the tubulin family and the functional features of centrosomes, cilia and flagella in various organisms. When reading the article, there is no doubt about the competence of the authors in matters of parasitology, biochemistry and molecular biology of the studied objects. Unfortunately, in the field of morphology, the article has significant shortcomings that need to be corrected before the article can be published in the journal Microorganisms. My special articles were devoted to correcting mistakes and inaccuracies made by the authors in the article, so it makes no sense to keep the reviewer anonymous in this case, since I recommend the authors to read my works.

We thank the reviewer for the corrections as they improve the paper.

Issue 1: Fig 1С. Line 67. I want to draw your attention to the fact that the direction of twisting (tilt) of triplets in centrioles is not arbitrary, but strictly determined. In your drawing, the child centriole is oriented toward the reader with its proximal end. In this case, the triplets should be shown twisted counterclockwise (in your drawing, it's the other way around). There is a special article on this issue.

Clockwise or anticlockwise? Turning the centriole triplets in the right direction!

Uzbekov R, Prigent C. FEBS Lett. 2007 Apr 3;581(7):1251-4. doi: 10.1016/j.febslet.2007.02.069. Epub 2007 Mar 6. PMID: 17368628 Free article. Review.

In addition, you write that you have depicted a mobile eyelash in Figure 1C: “Motile flagellar axonemes extend from basal bodies (blue) ...” and in fact the basal body of the motile cilium is not associated with the second (daughter) centriole. With the exception of the two central microtubules of the axoneme, your drawing shows the complex of the basal body and the primary (non-motile, sensory) cilium. See for more details:

Principal Postulates of Centrosomal Biology. Version 2020.

Uzbekov RE, Avidor-Reiss T. Cells. 2020 Sep 24;9(10):2156. doi: 10.3390/cells9102156. PMID: 32987651 Free PMC article. Review.

Thus, if you want to show the primary cilium in this picture, then you need to remove the central microtubules of the axoneme, and if you want to show the mobile cilium, then you need to remove the second centriole. Of course, the cilium-basal body complex is not limited to this simplified scheme. There are additional structures on the basal body (rootlets, foot...), but authors can give a more complete or simplified scheme. Main, so that there are no errors.

Response: We have revised figure 1C and hope that it now reflects the correct handedness of the daughter centriole. In order to capture information about both motile cilia and sensory cilia, we now illustrate both axoneme configurations. Similarly, the daughter centriole is now boxed off and we note that it is not associated with motile axonemes. The figure legend changes are highlighted in the revised version for reviewers.

Issue 2: Figure 1C gives an idea of the structure of the cilium-basal body complex in the cells of multicellular animals (more precisely, even in vertebrates). In various protozoans, which are the main objects of this study, the structure of this complex has its own characteristics, which the authors partially demonstrate in Figure 6. For example, two flagella in Chlamydomonas ... This discrepancy may cause readers to misunderstand some of the provisions expressed in the article. It is necessary to give more detailed schemes of the cilium-basal body complex in precisely those organisms to which the article is devoted.

Response: We agree and have tried to do this efficiently. See revised Figure 1 and new text for its figure legend (highlighted).

Issue 3: Line 86-87 Next, you write about appendages, in the diagram they are shown, but not indicated in any way. « Loss of δ- and ε-tubulins is associated with an inability to form appendage-containing basal bodies, or to build basal bodies and motile flagella [20-23].”

Response: We have labeled this in Fig 1C added a description of this in the new text for figure legend 1 (highlighted).

Issue 4: Line 284 “Most microtubules are anchored to microtubule organizing centers (MTOCs).” This is too categorical statement, because it is not true for most cell types. Most microtubules in the cytoplasm of cells are "free". They either broke away from the microtubule organization centers or were polymerized independently of them. Here it is necessary to indicate specifically those types of cells for which the authors are sure that their statement is true.

Response: We have tempered our language; our bias comes from working on an organism with stereotyped and highly defined microtubules. See highlighted text.

Issue 5: Fig. 6 Lines 405 and 411 and 420 and 478 and 484, 425, It is better to use the full formula reflecting the total number of microtubules in the axoneme  (9Ñ…2+2). 

Response: We have added this correction throughout. See highlighted text.

Issue 6: Line 444-447 «Although Drosophila sperm have motile flagella, their basal bodies lack subdistal appendages, structural extensions that typically stabilize the mature (mother) centriole/basal body and coordinate oriented connection to the plasma membrane [21, 79-81].» In the basal bodies of spermatozoa of any organism, as far as I know, there are no subdistal appendages ("foot", "arm"). The function of subdistal appendages is completely different in centrosomes. You probably mean here the analog of subdistal appendages in the basal bodies ("foot" in another terminology "arm" ). In the ciliary epithelium, they are indeed oriented identically for all cilia in the cell.

About subdistal appendages there is special paper:

Who are you, subdistal appendages of centriole?

Uzbekov R, Alieva I. Open Biol. 2018 Jul;8(7):180062. doi: 10.1098/rsob.180062. PMID: 30045886 Free PMC article. Review.

Response: Thanks for the clarification: we were trying to describe distinct combinations of outcomes for centrioles and basal bodies after the loss of delta and epsilon tubulin. We have modified this statement to “Drosophila centrioles lack subdistal appendages, structural elements that typically distinguish the mature (mother) centriole [21, 85-87]. These nine triplet containing structures are converted into basal bodies that template axonemes, including for motile flagella in sperm cells.” (highlighted).

Issue 7: Line 449. It is better to use the full formula reflecting the total number of microtubules in the axoneme (9x2+0)

Response: We have corrected this in the manuscript (highlighted).

Issue 8: The text of the paper a lot of word hyphenation from line to line. I would recommend authors to avoid this by using the justifier function.

Response: We agree! The manuscript in word does not have these hyphenations. They arise as a function of the specific format used by IJMS journals which is out of our control.

Reviewer 2 Report

Microtubules are cytoskeletal filaments essential for many cellular processes, including establishment and maintenance of polarity, intracellular transport, division and migration. In this study, the authors use bioinformatic approaches to analyze features of tubulins in organisms from the phylum Apicomplexa. The results obtained are novelty and informative, and provided some important clues for the subsequent biological studies. The analysis of the results is credible. The manuscript can be accepted for publishment after minor revise.

Major comments:

1.    The authors give a comprehensive review on construction and function of the microtubules in microgamete, however, a set of membrane-associated microtubules is essential for the elongated shape of invasive "zoites". The cortical microtubules are required to maintain the shape and rigidity of apicomplexan zoites, which have been proved in Plasmodium, Toxoplasma and Cryptosporidium parasites. How about the construction and function of membrane-associated microtubules? It will be better to add some information on the issue.

Minor comments:

1.    All species names should be italicized

2.    Fig4 C. reinhardtii , the abbreviation should be “Cs”, not “Cr”

Author Response

Please see attached highlighted MS for revisions

Reviewer 2: Microtubules are cytoskeletal filaments essential for many cellular processes, including establishment and maintenance of polarity, intracellular transport, division and migration. In this study, the authors use bioinformatic approaches to analyze features of tubulins in organisms from the phylum Apicomplexa. The results obtained are novelty and informative and provided some important clues for the subsequent biological studies. The analysis of the results is credible. The manuscript can be accepted for publishment after minor revise.

We thank the reviewer for the suggestion to define microtubule populations in apicomplexans as this request improves the clarity of the paper.

Major comments:

  1. The authors give a comprehensive review on construction and function of the microtubules in microgamete, however, a set of membrane-associated microtubules is essential for the elongated shape of invasive "zoites". The cortical microtubules are required to maintain the shape and rigidity of apicomplexan zoites, which have been proved in Plasmodium, Toxoplasma and Cryptosporidium parasites. How about the construction and function of membrane-associated microtubules? It will be better to add some information on the issue.

Response: Yes, this was an important missing piece. We have added a new paragraph to the introduction that described the various microtubule populations and their functions in apicomplexan parasites. This also permits us to include some information on the conoid, which also helps with the flow and content of the manuscript. See highlighted section.

Minor comments:

  1. All species names should be italicized

Response: We have checked for this in figures and text and do not identify any places where complete genus and species names are not italicized. Please cite any locations you have identified.

  1. Fig4 C. reinhardtii , the abbreviation should be “Cs”, not “Cr”

Response: We are not certain why this is requested: the initials for the accepted genus and species name are Cr, not Cs. It is possible that these is a second binomial name for the same species, but given that C. reinhardtii is the accepted name, we have left it in place.

Round 2

Reviewer 1 Report

Dear Editor
In my opinion, a corrected version of the article can be published now.